# Age-Dependent Hippocampal Proteomics in the APP/PS1 Alzheimer Mouse Model: A Comparative Analysis with Classical SWATH/DIA and directDIA Approaches

**DOI:** 10.3390/cells10071588

**Published:** 2021-06-24

**Authors:** Sophie J. F. van der Spek, Miguel A. Gonzalez-Lozano, Frank Koopmans, Suzanne S. M. Miedema, Iryna Paliukhovich, August B. Smit, Ka Wan Li

**Affiliations:** Center for Neurogenomics and Cognitive Research, Department of Molecular and Cellular Neurobiology, Amsterdam Neuroscience, Vrije Universiteit Amsterdam, 1081 HV Amsterdam, The Netherlands; s.j.f.vander.spek@vu.nl (S.J.F.v.d.S.); m.a.gonzalezlozano@vu.nl (M.A.G.-L.); frank.koopmans@vu.nl (F.K.); s.s.m.miedema@vu.nl (S.S.M.M.); i.paliukhovich@vu.nl (I.P.); guus.smit@vu.nl (A.B.S.)

**Keywords:** APP/PS1 mouse model, Alzheimer, microglia, proteomics, directDIA, MS-DAP

## Abstract

Alzheimer’s disease (AD) is the most common neurodegenerative disorder in the human population, for which there is currently no cure. The cause of AD is unknown; however, the toxic effects of amyloid-β (Aβ) are believed to play a role in its onset. To investigate this, we examined changes in global protein levels in a hippocampal synaptosome fraction of the Amyloid Precursor Protein swe/Presenelin 1 dE9 (APP/PS1) mouse model of AD at 6 and 12 months of age (moa). Data independent acquisition (DIA), or Sequential Window Acquisition of all THeoretical fragment-ion (SWATH), was used for a quantitative label-free proteomics analysis. We first assessed the usefulness of a recently improved directDIA workflow as an alternative to conventional DIA data analysis using a project-specific spectral library. Subsequently, we applied directDIA to the 6- and 12-moa APP/PS1 datasets and applied the Mass Spectrometry Downstream Analysis Pipeline (MS-DAP) for differential expression analysis and candidate discovery. We observed most regulation at 12-moa, in particular of proteins involved in Aβ homeostasis and microglial-dependent processes, like synaptic pruning and the immune response, such as APOE, CLU and C1QA-C. All proteomics data are available via ProteomeXchange with identifier PXD025777.

## 1. Introduction

Alzheimer’s disease (AD) is the most common age-related neurodegenerative disorder. There is currently no cure, and available medical treatments aim at alleviating symptoms. While the cause of AD is under investigation, accumulation of amyloid-β (Aβ) derived from the aberrant proteolytic cleavage of the amyloid precursor protein by γ- and β-secretases is believed to play an important role in its pathogenesis [1,2]. Deposition of Aβ progresses slowly throughout the brain, originating in basal cortical areas, spreading through the hippocampus and ultimately affecting all areas of the cortex [3]. Additional pathological changes in the brains of AD patients include the presence of neurofibrillary tau tangles, astro- and microgliosis, synapse loss and neuronal death [3].

*APP*swe*/PSEN1*dE9 transgenic mice [4] express two human pathologically mutated genes, the Amyloid Precursor Protein swe and Presenilin 1 dE9 (further mentioned as APP/PS1). The APP/PS1 mouse model is one of the most widely used to study AD [5] and specifically recapitulates the amyloid production aspect of the disease. For example, APP/PS1 mice show early elevated Aβ production and plaque formation in the hippocampus observed at 6 months of age (moa), which increases progressively with age [6]. In addition, these mice show synaptic dysfunction [7,8], presence of reactive astrocytes [9] and microglia [10], and multiple forms of memory impairments [8,11]. Studies have indicated synapse loss as early events in AD, which especially affect the hippocampus, and correlates with cognitive decline [3,12]. Aβ exists in multiple forms and appears to play an important role in synapse toxicity [3]. As synapse dysfunction and/or loss are believed to underlie the early pathology of AD, it is necessary to unravel the temporal changes in molecular and cellular processes encompassing the synapse in relation to the advancing Aβ challenge. To this end, we performed proteomics analysis on hippocampal synapse enriched fractions obtained from 6- and 12-moa APP/PS1 mice and their wildtype controls.

Alterations of molecular and cellular processes can be inferred from changes in protein expression levels. Proteomics technology, capable of quantifying thousands of proteins from a small sample size, is the method of choice to shed light on this. Data independent acquisition (DIA), also referred to as Sequential Window Acquisition of all THeoretical fragment-ion (SWATH) [13], is a quantitative proteomics methodology that offers quantification of a high number of proteins with low technical variation and number of missing values. Classical SWATH analysis requires a project-specific spectral library generated from separate data-dependent acquisition (DDA) runs for protein identification. A recently improved directDIA workflow now enables the creation of a spectral library directly from the SWATH samples, bypassing the need for additional mass spec run-time and sample fractionation [14,15].

In the present study, we used SWATH to examine differential protein expression levels in hippocampal synaptosomes of 6- and 12-moa APP/PS1 mice. First, we compared the use of directDIA versus a project-specific spectral library, and we revealed that directDIA preferentially identifies high-intensity peptides, consistently across all sample replicates, resulting in high data completeness. Importantly, using either method resulted in the detection of a largely overlapping group of significantly regulated proteins, which, together, validates the directDIA workflow as a good alternative to the use of a DDA library. Subsequently, we applied directDIA to both 6- and 12-moa datasets and used the Mass Spectrometry Downstream Analysis Pipeline (MS-DAP) for differential expression analysis and protein candidate discovery. We observed upregulation of multiple AD-associated proteins in the 12-moa APP/PS1 mice. At 6 months, these proteins revealed no regulation or lower levels of regulation. Proteins most differentially regulated between the two ages were those that are implicated in Aβ homeostasis and microglial-dependent synaptic pruning and/or immune activation such as APOE, CLU and C1QA-C.

## 2. Materials and Methods

### 2.1. Mice

The use of APP/PS1 mice in this study was approved by the animal ethical care committee of the Vrije Universiteit Amsterdam. All wildtype and APP/PS1 mice of 12-moa were males. Both conditional groups at 6-moa were a mix of males and females (3 of each sex/condition).

### 2.2. Synaptosomal Enrichment

For all age and genotype conditions, synaptosomal fractions of 5 or 6 mice were individually prepared and analyzed. Samples were prepared as previously described [14,16]. Mouse hippocampi were dissected and stored at −80 °C until further use. Per mouse, the two hippocampi from both hemispheres were homogenized together in 6 mL of homogenization buffer (0.32 M sucrose (VWR, Radnor, PA, USA), 5 mM HEPES (Sigma-Aldrich, St. Louis, MO, USA) pH 7.4, Protease inhibitor cocktail (Roche, Basel, Switzerland)). Samples were homogenized using a potter and pestle (Sartorius, Göttingen, Germany; 12 strokes, 900 rpm) and centrifuged at 1000× *g* for 10 min at 4 °C. Subsequently, 4.5 mL of supernatant was loaded on top of a 0.85/1.2 M (6 mL each) sucrose gradient and centrifuged at 100,000× *g* for 2 h. Per sample, 1.5 mL synaptosomes were recovered between 0.85/1.2 M sucrose interface, mixed with 3.5 mL 5 mM HEPES, pH 7.4, and centrifuged at 20,000× *g* for 30 min to obtain the synaptosomal pellets. Synaptosomes were resuspended in 150 μL homogenization buffer, and protein concentration was determined with a Bradford assay (Protein Assay, Bio-rad, Hercules, CA, USA). 

### 2.3. Filter-Aided Sample Preparation

Samples were digested following the filter-aided sample digestion protocol [17] with some modifications. In short, for each sample, 22 μg synaptosomes were solubilized in 100 μL 2% sodium dodecyl sulfate (Sigma-Aldrich, St. Louis, MO, USA) containing 1 μL 500 mM Tris (2-carboxyethyl) phosphine (Sigma-Aldrich, St. Louis, MO, USA) reducing reagent, at 55 °C for 1 h. Next, cysteine residues were blocked with 0.5 μL 500 mM methyl methanethiosulfonate (Fluka, Honeywell, Charlotte, NC, USA) for 15 min at room temperature. After addition of 200 μL 8M urea (Sigma-Aldrich, St. Louis, MO, USA) in tris buffer (Sigma-Aldrich, St. Louis, MO, USA), pH 8.8, the samples were transferred to YM-30 filters (Microcon^®^, Millipore, Burlington, MA, USA) and centrifuged at 14,000× *g* for 15 min. The samples were washed with 8 M urea solution four times by centrifugation at 13,500× g for 14 min each, followed by four washes with 50 mM ammonium bicarbonate (Sigma-Aldrich, St. Louis, MO, USA). Trypsin (Mass Spec Grade, Promega, Madison, WI, USA; 0.6-g trypsin in 100 μL 50 mM ammonium bicarbonate) was added to the proteins on filter and incubated overnight at 37 °C. The filters were centrifuged, and the digested peptides were collected in a clean centrifuge tube. The samples were dried in a speedvac (Savant, Thermo Scientific, Waltham, MA, USA) and stored at −20 °C until Liquid Chromatography–Tandem Mass Spectrometry (LC-MS/MS) analysis.

### 2.4. Micro-LC and SWATH Mass Spectrometry

Peptides were analyzed by micro-LC MS/MS using an Ultimate 3000 LC system (Dionex, Thermo Scientific, Waltham, MA, USA) coupled to the TripleTOF 5600 mass spectrometer (Sciex, Framingham, MA, USA) as described previously [16,18,19]. Peptides were trapped on a 5 mm Pepmap 100 C18 column (Dionex, Thermo Scientific, Waltham, MA, USA; 300 μm i.d., 5 μm particle size) and fractionated on a ChromXP C18 column (Eksigent, Sciex, Framingham, MA, USA; 3 μm particle size, 120A). The acetonitrile (VWR, Radnor, PA, USA) concentration in the mobile phase was increased from 5 to 18% in 88 min, to 25% at 98 min, 40% at 108 min and to 90% in 2 min, at a flow rate of 5 μL/min. The eluted peptides were electro-sprayed into the TripleTOF 5600 mass spectrometer, with a micro-spray needle voltage of 5500 V. SWATH experiments consisted of a parent ion scan of 150 ms followed by a SWATH window of 8 Da with scan time of 80 ms, that stepped through the mass range between 450 and 770 *m/z*. The collision energy for each window was determined based on the appropriate collision energy for a 2+ ion, centered upon the window with a spread of 15 eV. 

### 2.5. SWATH Data Analysis 

Spectronaut 14 (Biognosys, Schlieren, Switzerland) was used for data analysis of the raw files. All SWATH runs of the 12-moa experimental sample set were analyzed against both the spectral library and an internal spectral library using the directDIA function in Spectronaut 14. Analysis against the spectral library was done in the Analysis Perspective of Spectronaut by uploading all raw files, assigning the spectral library to each file and applying the Biognosys (BGS) Factory Settings. Analysis using the directDIA function in the Analysis Perspective of Spectronaut was performed by uploading raw files and assigning the mouse reference proteome files (the 2018_04 Uniprot release of UP000000589_10090.fasta and UP000000589_10090.additional.fasta). Additionally here, the Biognosys Factory Settings were applied. The 6-moa runs were analyzed only using directDIA, the same way as the 12-moa dataset. Before exporting data from Spectronaut, all filters were disabled. The dedicated spectral library was created with crude hippocampal synaptosomes containing spiked-in indexed Retention Time peptides (Biognosys, Schlieren, Switzerland), analyzed with the Triple TOF 5600 in DDA mode. The obtained library data were searched against the mouse proteome (the 2018_04 Uniprot release of UP000000589_10090.fasta and UP000000589_10090.additional.fasta) in Maxquant. Methyl methanethiosulfonate (C) was set as fixed modification. In the Library Perspective of Spectronaut, the dedicated spectral library was generated by uploading the Maxquant evidence.txt and modifications.xml files, the used fasta files, and assignment of the Shotgun Files (raw files). 

The Mass Spectrometry Downstream Analysis Pipeline (MS-DAP) (available at https://github.com/ftwkoopmans/msdap; version beta 0.2.5.1) was used for quality control and candidate discovery. In MS-DAP, peptide intensities without normalization in Spectronaut were taken for downstream analysis. For differential expression analysis, the 6- and 12-moa datasets were analyzed separately. Peptides present in at least 75% of the wildtype or APP/PS1 groups were used for differential testing, with the limma emperical Bayes algorithm after rollup to proteins. Shared peptides were removed, and the Variation Within Mode Between and modebetween_protein algorithms were used for normalization. All proteomics data used here have been deposited to the ProteomeXchange Consortium via the PRIDE [20] partner repository with the dataset identifier PXD025777.

## 3. Results

We first investigated the enrichment of synaptic proteins in the synaptosomal sample preparation and suitability of directDIA to reveal the effects of APP/PS1 Aβ expression on synaptic protein levels in animals at 6 and 12 moa. Synaptosomes were isolated using a standard protocol that in the past showed high reproducibility [21,22,23,24]. We performed GO-enrichment analysis on all proteins identified in the synaptosomal preparations under investigation in the current study. Using total brain genome as background, this revealed ‘synapse’ as strongest enriched term in both 6- and 12-moa datasets (Appendix A). 

We continued investigating the suitability of directDIA for the analysis of our datasets. Label-free quantification mass spectrometry can be performed in DDA or DIA/SWATH mode. The classic approach of SWATH analysis uses a spectral library generated from extensive DDA analysis for protein identification. Recent developments in data analysis enable the construction of a library directly from the SWATH data in a workflow called directDIA [14,15]. Here we performed SWATH analysis and used the new software suite Spectronaut 14 containing the directDIA (2.0) workflow. 

For comparison, we first searched the 12-moa APP/PS1 dataset (*n* = 6/condition) against our standard in-house hippocampal DDA-based spectral library in Spectronaut for peptide and protein identification and quantification, and we ran the data through MS-DAP, a recently released downstream analysis pipeline for quantitative proteomics (available at https://github.com/ftwkoopmans/msdap; version beta 0.2.5.1). This newly developed all-in-one analysis tool provided extensive quality control plots, allowed filtering and normalization of data, and revealed significantly changed proteins between experimental conditions by differential testing. The analysis resulted in the detection of 31,670 peptide precursors on average per sample. A sizeable fraction of peptides fell below the 0.01 confidence threshold and represents potential false positives (Figure 1a; Appendix A). Filtering out the low-quality precursors with a *q*-value > 0.01 removed, on average, 11,058 precursors (35%) per sample. On average, 20,612 (65%) precursors were retained per sample that were mapped to 19,413 target peptides and 3374 proteins (Appendix A). A total of 15,306 peptides were quantified in all samples (Figure 1b).

To assess the performance of directDIA for analysis, we generated an internal spectral library with the same 12-moa dataset using the directDIA feature in Spectronaut 14. Using the library from directDIA, we detected on average 22,600 precursor peptides per sample. When the precursors were filtered on quality, per sample, an average of 4972 precursors (22%) with a *q*-value > 0.01 were removed, and 17,628 (78%) identifications were retained. These retained precursor identifications mapped to an average of 12,473 unique peptides and 2304 proteins per sample (Appendix A). This is 36% and 32% fewer peptides and proteins than observed in the search against the project-specific spectral library. Against this apparent disadvantage of directDIA, the confidence score distributions clearly showed that the relative number of potential false-positive identifications, and loss of identifications after filtering for quality, was much lower using directDIA (Figure 1c; Appendix A). Of interest, the use of directDIA results in a high data completeness with nearly all identified peptides (98%) consistently observed across all sample replicates (Figure 1d). 

Using directDIA a total of 12,517 peptides with a *q*-value ≤ 0.01 were identified in the entire 12-moa dataset, and 23,061 using the spectral library (Appendix A). The large number of peptides uniquely identified using the spectral library (11,574) (Appendix A) were of lower intensity (Appendix A) and quality (Appendix A) than the peptides identified in both the spectral library and directDIA (11,487). The intensities of peptides shared by both searches showed a high correlation (R^2^ = 0.95) (Appendix A), suggesting these peptides are based on the same peaks, and both workflows perform, to a large extent, equally. 

To compare the effects of peptide identification using the DDA-based spectral library or directDIA on further downstream analysis, we first ran a differential expression analysis (DEA) on the 12-moa APP/PS1 experimental group and their wildtype controls searched against both types of libraries. For each search, DEA was performed using peptides detected with a *q*-value ≤ 0.01 in at least 75% of the samples in each experimental group. In addition, peptides shared between proteins were removed. DEA using the DDA-based spectral library and directDIA was performed on 17,153 peptides that mapped to 3039 proteins, and 12,441 peptides mapped to 2300 proteins, respectively (Figure 2a). Most proteins retained for DEA were observed in both dataset searches (74%) (Figure 2b). Those proteins observed only using the DDA-based spectral library were, for the largest part, based on one peptide (67%) (Figure 2c) and, as expected based on the earlier peptide analysis, of lower abundance (Appendix A).

Differential testing of protein expression levels revealed 23 and 32 regulated proteins with statistical significance (empirical Bayes corrected *p*-values, or *q*-values ≤ 0.01) in APP/PS1 mice versus their wildtype controls using the DDA-based spectral library search and directDIA, respectively (Figure 2d). Importantly, the majority of these proteins (21) were found significantly altered within both searches (Figure 2d). Of the two proteins detected as regulated uniquely in the DDA-based spectral library searched dataset, one showed significance with directDIA search when relaxing the criteria to a *q*-value ≤ 0.05 (Figure 2d). Relaxing the criteria to a *q*-value ≤ 0.05 revealed a substantial number of proteins reaching significance only in the DDA-based spectral library (49) or the directDIA library (33) (Figure 2d). Of the 49 significant proteins observed only using the DDA-based spectral library, 16 were not detected with directDIA, and the additional 33 revealed higher fold-changes than observed using directDIA (0.20 ± 0.11 versus 0.15 ± 0.1 log_2_ fold-change, respectively), which is therefore likely the cause of reaching significance at a *q*-value ≤ 0.05 using the DDA-based spectral library only. Variation in abundance for these proteins was the same between the two libraries (0.15 ± 0.08 versus 0.15 ± 0.1 SD, respectively). Of the 33 proteins only found significant using directDIA, two were not identified using the DDA-based spectral library. The additional 31 proteins revealed lower variation than observed with the DDA-based spectral library (0.15 ± 0.06 versus 0.27 ± 0.48 SD, respectively), while revealing the same log_2_ fold-changes (0.18 ± 0.16 and 0.18 ± 0.17, respectively). As directDIA gives high confident peptides, and showed a higher number of significant proteins at a *q*-value ≤ 0.01, we proceeded with directDIA for our 6-moa dataset in a subsequent analysis.

In the 6-moa dataset (*n* = 6/condition), quality control using MS-DAP showed a clear outlier in the wildtype group possibly due to a technical issue of the high-performance liquid chromatography run, and was removed from further analysis (Appendix A). Like the 12- moa dataset run with directDIA, the 6-moa dataset showed high data completeness across samples (Appendix A), and similar numbers of peptides and proteins were detected after filtering for DEA (12,010 and 2469 on average per sample, respectively) (Appendix A). Both 6- and 12-moa datasets showed low coefficients of variation, ranging between 8 and 12.3%, per experimental condition (Appendix A), and the majority of proteins used for DEA were detected and tested in both age groups (78%) (Appendix A).

In the 6-moa dataset, only a few proteins showed significant regulation with a *q*-value ≤ 0.01 (Figure 3a). These included NCSTN (one of the subunits of the gamma-secretase PS1 complex), DOCK9 and TXNRD1 that had higher levels in APP/PS1 mice, and GPC4 that showed a decrease in expression (Figure 3b). The level of APP itself was also increased, but at a higher *p*-value of 0.026 (Figure 3b). In contrast, the 12-moa dataset showed 30 proteins up and 2 down (MTMR1 and GAK) in the APP/PS1 group (Figure 3a). At 12-moa APP showed an increased level, and similar to the 6-moa dataset, a strong increase was observed for NCSTN and DOCK9 (Figure 3b). Besides APP and NCSTN, several of the additional most regulated proteins are known AD risk factors or proteins related to AD pathology, including APOE, CLU and C1QA-C (Figure 3b, Appendix A). Indeed, Gene Ontology enrichment analysis in gProfiler using the mouse proteome as background revealed multiple significant terms associated with AD, including “regulation of amyloid fibril formation” (Biological Process) and “Alzheimer’s disease” (WikiPathways) (Figure 4a). Using our custom total list of proteins detected at 12-moa as background, Gene Ontology analysis showed enrichment of terms such as “Membrane proteolysis” (Biological Process) and “lysosome” (Cellular Component) (Figure 4b). Both terms were largely based on APP, APOE and NCSTN. In addition, the lysosomal term included VTl1B, ARL8B, EPDR1, HEXB, SYT11 and the AD-associated proteins PLD3 and LAMP2 (Appendix A). An additional known AD-protein not annotated to Aβ, Alzheimer or lysosome-related terms, which was regulated here, was AQP4 (Figure 4b, Appendix A).

To reveal possible specific subsynaptic compartments affected in APP/PS1 mice, we performed enrichment analysis with the Synaptic Gene Ontology (SynGO) knowledge base [25]. The list of 34 regulated proteins at 6- or 12-moa contained 11 proteins that were annotated to SynGO (Appendix A). The additional proteins were not annotated in SynGO yet, or they came from non-synaptic impurities of the synaptosomal sample preparation with similar biochemical properties. The regulated synaptic proteins were equally annotated to the pre- and post-synapse (Appendix A), without significant enrichment towards a specific subsynaptic localization or function (Appendix A).

We then performed an Expression Weighted Cell-type Enrichment (EWCE) analysis [26] on the significantly upregulated proteins at 12-moa to distinguish the possible contributions of different cell types to the changed expression levels. This is based on the notion that these proteins will be non-equally expressed over all cell types. For EWCE analysis we used previously published single-cell RNAseq gene expression profiles obtained from mouse hippocampus [27]. We observed overrepresentation of microglial and, to a lesser extent, astrocytic proteins, albeit not significant (Figure 5a), which are therefore likely the major source of the observed increase in protein expression. Proteins with high microglial expression include C1QA-C and HEXB, and proteins showing high astrocytic expression include AQP4, GPC4, CLU and ATP1A2 (Appendix A).

The level of APP expression showed an increase over time, and additional protein regulations were also stronger at 12-moa than at 6-moa (Figure 3b). To visualize the progressive temporal changes in expression of regulated proteins directly, we derived the fold change ratios of proteins significant in at least one of the two datasets (Figure 5b). Here we observed that the top 10 of proteins with the highest changing expression levels over time include APP, C1QB/C, APOE, CLU and AQP4 (Figure 5b). 

## 4. Discussion

Toxicity and accumulation of Aβ is believed to play important roles in AD pathogenesis [1,2], starting in basal cortical areas spreading through the hippocampus and other areas of the cortex [3]. In addition, synapse loss has been indicated as early events in AD that correlate strongly with cognitive impairment [3,12]. Studies have shown that Aβ is important for synaptic failure [3]. As synapse loss especially affects the hippocampus [12], this structure was our brain area of interest. 

Because aberrant molecular and cellular changes in and around the synapse are considered to play a part in the cause of AD progression, it is necessary to unravel their temporal changes in relation to the advancing Aβ challenge. To reveal changes in protein expression that may result from the overexpression and aberrant processing of APP into Aβ, we examined the hippocampal proteome of 6- and 12-moa APP/PS1 mice and their wildtype controls. For our study we employed SWATH technology [13]. Current studies indicate that SWATH yields small variations and few missing values among samples that together enable the detection of subtle changes in expression, and may therefore be the preferred method of choice [16,19].

A typical SWATH experiment requires a project-specific spectral library for peptide identification during a database search. Building a spectral library requires extensive DDA analysis, preferably on the same or similar samples, and measured under comparable conditions to the measurements done on the samples of interest. This increases measurement time, and the conditions to generate the library are often not an exact copy of those while measuring the samples, leading to an increased chance of spectra mismatching. In contrast, directDIA assembles the precursor ions and fragment ions into pseudo–tandem Mass Spectrometry spectra, which can be built into a spectral library by the search against a conventional reference proteome database. The project-specific spectral library search in this study produced more peptides than did a directDIA library search. However, among the uniquely detected peptides, many of these were of lower intensity and quality compared to those identified by both workflows. These peptides are likely causing the reduced consistency observed in peptide detection across the sample replicates. Within the shared peptides generated from directDIA and the project-specific spectral library we revealed a high correlation in peptide intensity. This suggests these peptides are based on the same peaks and implicates that both approaches quantified peptides in similar ways. Importantly, downstream analysis showed that the directDIA protocol detected more significantly regulated proteins with high confidence, in addition to the shared proteins, demonstrating the usefulness of directDIA for database search.

We continued using directDIA for the analysis of protein changes in APP/PS1 mouse hippocampal synaptosomes of 6- and 12-moa. Synaptosomes were isolated with a standard protocol, which in the past has shown enrichment of synaptic proteins with high reproducibility [21,22,23,24]. Additionally, in the synaptosomal preparation under investigation, we observed enrichment of synaptic proteins, as revealed by GO-enrichment analysis. In addition, the synaptosomal fraction may contain structures with similar biochemical properties or structures of contacting non-neuronal cells. For instance, microglia have been implicated in the elimination of synapses during development and under pathological conditions such as exposure to Aβ oligomers [28]. Synapse engulfment and pruning by microglia occurs in a complement factor dependent manner [28,29]. Indeed, highest regulated proteins observed in our dataset are strongly expressed in microglia, as reflected in the cell-type enrichment analysis, and include the complement factors C1QA-C. 

Despite enriching for synaptic proteins, surprisingly, only a few synaptic proteins are regulated in the APP/PS1 mice at 6- and 12-moa (e.g., SYT11 and GABRB3). SynGO-analysis revealed no enrichment of these proteins towards a specific subsynaptic compartment or biological process. In contrast, a recent proteomics study on the human (pre-clinical) AD cortex revealed changes in proteins related to the secretory pathway and synaptic vesicle endocytosis (e.g., SYT2 and SH3SGL2) [30], supporting the relevance of synaptic homeostasis in AD disease pathology. Proteins related to these pathways were among early responding, late responding and progressively changing protein groups [30]. In the APP/PS1 mouse model of AD, stronger changes in synaptic vesicle endocytosis proteins have been observed at 3-moa [4,31], suggesting the APP/PS1 model recapitulates especially early synaptic changes induced by Aβ. This is in line with a recent cross-species meta-analysis on human AD transcriptomics and mouse models of AD [32]. Among different human AD studies, the meta-analysis revealed consistent downregulation of gene groups enriched for neuronal genes [32]. The strongest overlap of regulated neuronal genes was observed in mice with a mild pathological burden [32].

In the current study, strongest (up-)regulation of proteins was observed in the 12-moa dataset and contained multiple microglial proteins. At 6-moa we observed no regulation of microglial proteins. In contrast, microglial activation has been observed in multiple amyloidosis mouse models of AD as one of the earliest phenotypes [33,34,35]. For example, a recent transcriptomics study on microglial cells enriched from *App*^NL-G-F^ mice [36] revealed upregulation of microglia in an activated state already at 3-moa [35]. This activated group of microglia was characterized by increased expression of *Apoe* [33,34,35], an apolipoprotein involved in lipoprotein homeostasis and clearance of Aβ [37,38]. Removal of *Apoe* resulted in suppression of microglial recruitment to Aβ-plaques, revealing a pivotal role for this transcript in Aβ-related microglial activity [35]. At 12-moa, we also observed increased levels of APOE and other proteins previously detected in activated microglia, including HEXB, EPDR1, ERP29, RER1, GLG1 and PLD3 [33,35].

Lack of observed microglial protein regulation at 6-moa in the current study may be caused by several aspects. First, previous transcriptomics and proteomics studies isolated microglia before protein or transcript extraction [33,34,35]. As synaptosomes were used in the current study, there is reduced sensitivity towards changes that occur in this specific cell type. For example, APOE is expressed in both microglia and astrocytes, but *Apoe* mRNA was particularly shown to increase in microglia upon exposure to Aβ [35]. More subtle increases of microglial APOE at earlier timepoints may remain undetected due to dilution of protein changes. In addition, strong, early microglial effects were observed using amyloidosis mouse models, different from the APP/PS1 model used here [33,34,35]. Of interest, in *App*^NL-G-F^ mice [36] the relative number of reactive microglia increased over time from 6% at 3-moa, to 33% at 6-moa and 52% at 12-moa [35]. The same report also revealed an increase in reactive microglia in the APP/PS1 mouse model [4,35], the same model that was used in our current proteomics study [4]. This confirms activation of microglia as a consistent phenotype [35]. However, the APP/PS1 mouse only revealed a 15% increase at 18-moa [35], suggesting activation of microglia in different models follows distinct timelines or differences in severity. 

A difference in microglia activation rate may be due to differences in Aβ pathophysiology between amyloidosis mouse models, as suggested in a previous report [33]. In a recent study, APP/PS1 mice (bearing the *APP*swe and *PSEN1*L166P mutations) [39] were shown to contain fibrillar Aβ plaque cores at 3-moa, along with protein abundancy alterations in isolated microglia [33]. In contrast, fibrillar Aβ was barely detectable in *App*^NL-G-F^ mice [36] of the same age with similar plaque load, and showed no microglia proteome alterations [33]. At higher age, both models expressed dense core fibrillar Aβ and an altered microglial proteome [33]. In line with this, the cross-species meta-analysis showed upregulation of human AD gene groups enriched for microglial genes, most strongly and consistently in mice with severe Aβ pathology [32]. 

Proteins regulated in the APP/PS1 datasets observed in the current study were over-represented by those involved in APP processing and Aβ formation. For example, NCSTN is an integral component of the γ-secretase complex comprising PS1-NCSTN-APH1-PSENEN, which cleaves APP to produce Aβ peptides [40]. Upregulation of NCSTN suggests increased levels of γ-secretase, likely due to overexpression of the *PSEN1* gene in the APP/PS1 mouse model. PS1, APH1 and PSENEN were not detected in the proteomics dataset. The increased expression of APP is also most likely the direct consequence of the over-expression of the human *APP* gene. 

RER1 also showed elevated levels in the APP/PS1 mice. This protein regulates the retrieval of endoplasmic reticulum membrane proteins from the early Golgi compartment [41]. Correspondingly, RER1 was revealed to affect γ-secretase assembly by regulating retention and retrieval of NCSTN and PSENEN [42,43,44]. Increased levels of RER1 expression cause reduced maturation of APP, negatively regulating the production of Aβ [44]. This suggests in the APP/PS1 mice, elevated levels of RER1 work to compensate for increased APP and NCSTN levels.

APOE and CLU (APOJ) were highly up-regulated in our dataset, and they are well-established genetic risk factors of late-onset AD [45]. Although these are two distinct proteins, they show many similarities. Both APOE and CLU are apolipoproteins mediating lipid transport between cells in the brain [37,46]. These proteins are mainly secreted by astrocytes, in healthy brains, and are associated with immune modulation including activation of microglia [47], and are possibly involved in microglia-associated phagocytosis of Aβ [38]. Thus, increased levels of APOE and CLU in the APP/PS1 dataset may reflect cellular responses towards increased Aβ clearance.

GO analysis revealed additional enriched terms including “Complement Activation, Classical Pathway” as well as “synapse pruning”. These are based on C1QA-C, three complement factors that initiate the classical complement immune response [48]. These proteins are highly expressed by microglia and, together with other complement factors, are found in human and mouse Aβ plaques [49,50]. Increasing evidence shows a detrimental role of C1Q in AD pathogenesis, as it can enhance Aβ fibrillogenesis [51,52,53], block Aβ uptake by microglia [54] and are involved in aberrant synaptic pruning [28]. Thus, C1Q upregulation observed here likely contributes to AD pathology.

In the APP/PS1 mouse we also observed GO enrichment of proteins expressed in the lysosome, for instance HEXB and LAMP2. *Lamp2* revealed high expression in microglia in the single-cell RNAseq data [27]. In addition, *Pld3* transcripts and HEXB revealed enriched expression in reactive microglia in AD mice [33,35]. Microglia may be the main source of increased lysosomal proteins observed in the APP/PS1 dataset. Of interest, an AD study on the microglial proteome revealed enrichment of phagocytic and lysosomal proteins alongside impaired microglial phagocytotic capabilities [33]. As the authors suggested, increased phagocytic and lysosomal protein expression in microglia may be part of a compensatory mechanism to enhance microglial phagocytosis of Aβ. This response eventually fails to improve capabilities for the removal of Aβ [33]. 

Taken together, regulation of proteins involved in APP and Aβ processing (NCSTN, APOE, CLU and RER1), microglial activity (C1QA-C, APOE, HEXB, PLD3, LAMP2, EPDR1, ERP29, RER1 and GLG1 and PLD3) and the endo-lysosome (PLD3, VTI1B, EPDR1, HEXB, ARL8B and LAMP2) in this APP/PS1 mouse model reflect multiple aspects of AD-related processes. Regulation of these proteins in the APP/PS1 mouse model reinforces their importance in Aβ-induced pathology.

In addition, we observed proteins not linked to AD previously. Several of these have been associated with other neurodegenerative disorders (ZDHHC17 with Huntington’s disease [55], SYT11 and GAK with Parkinson’s disease [56] and GABRB3 with dementia with Lewy bodies [57]), and may be of special interest for future studies. We also detected dysregulated proteins with no reported relation to AD or neurodegeneration (e.g., AKAP or PIP4P2). Of special interest is DOCK9, which shows significant and high upregulation at both 6- and 12-moa. This suggests a role of DOCK9 as early responder to increased Aβ levels or participation of this protein in the production of Aβ. DOCK9 is a guanine nucleotide-exchange factor (GEF) that activates CDC42, a small effector protein involved in variety of cellular responses including cell migration [58]. Of interest, CDC42 activity has recently been shown to facilitate the microglial migration response to Aβ, downstream of TREM2 [58]. TREM2 is a receptor expressed by microglia and is a risk factor for AD [59]. Although speculative, DOCK9 may be involved in similar migratory pathways. In addition, family members DOCK2 and 3 have shown involvement in the regulation of Aβ plaque load [60] and phosphorylation of tau [61], respectively, making DOCK9 an interesting candidate for future studies.

## Figures and Tables

**Figure 1 cells-10-01588-f001:**
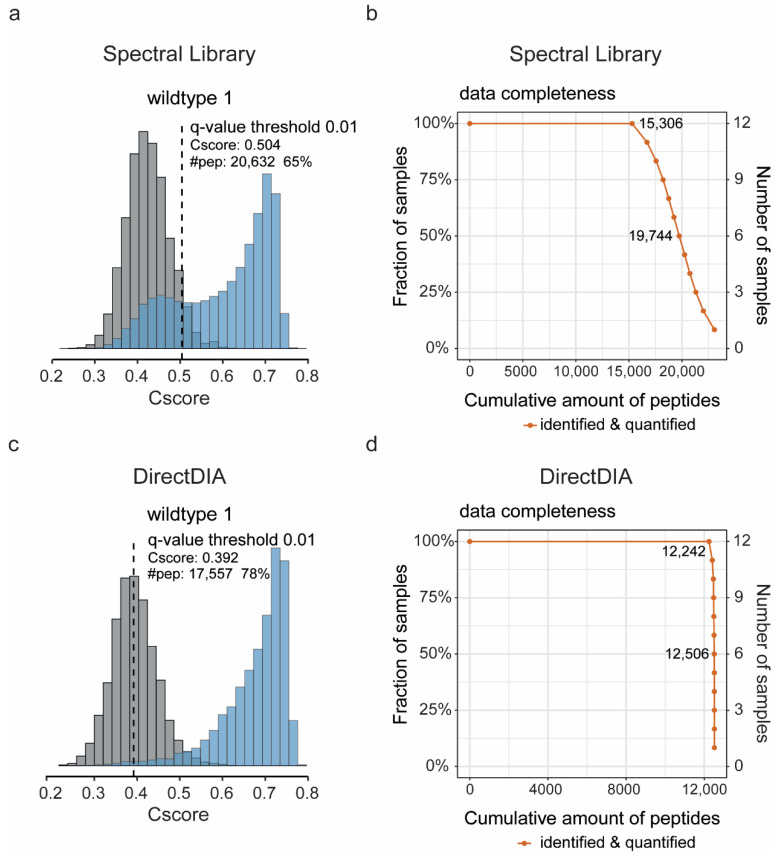
Comparison between the use of a Data Dependent Acquisition (DDA)-based spectral library or directDIA for protein identification and quantification. (**a**) Exemplary histogram of one sample (wildtype 1) showing both target (blue) and decoy (grey) confidence scores (cscores), indicating the level of confidence of peptide identification using the DDA-based spectral library. The *q*-value confidence threshold of 0.01 is shown as a dotted line, and the associated cscore and number of peptides quantified above this threshold are reported. (**b**) Cumulative distribution showing the number of peptides consistently identified across the range of samples, using the DDA-based spectral library. The exact number of peptides identified are shown at 100% or 50% of samples. (**c**) Analogous to panels a and b: an exemplary histogram of the same sample is shown to visualize target and decoy cscores obtained after identification of peptides in the raw SWATH data using directDIA. (**d**) A cumulative distribution showing the number of peptides consistently identified across the samples, using directDIA. Cscore histograms of all individual samples run against the DDA-based spectral library or directDIA library are reported in Appendix A, respectively.

**Figure 2 cells-10-01588-f002:**
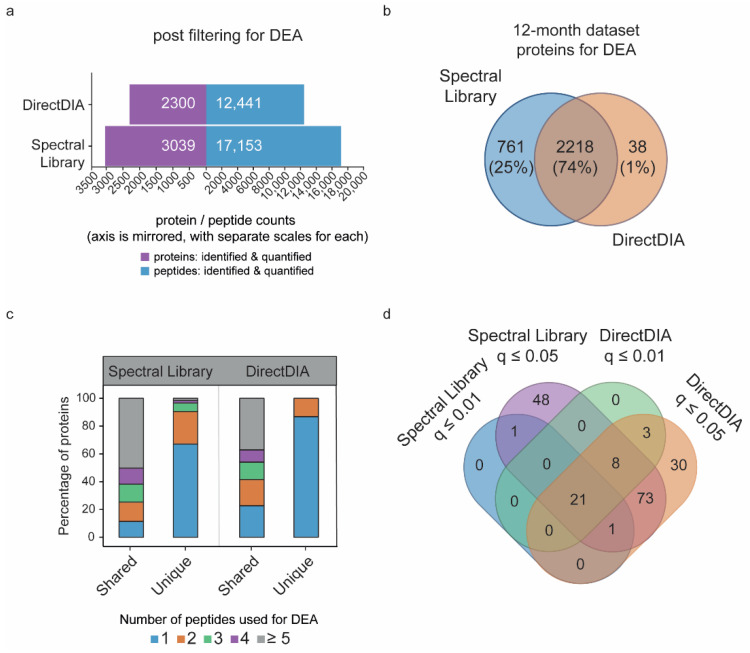
Characterization of proteins used for differential expression analysis in MS-DAP obtained by a DDA-based spectral library or directDIA. (**a**) Number of proteins and peptides that remain after filtering for differential expression analysis. (**b**) Number of unique and shared proteins used for downstream analysis identified by the DDA-based spectral library or directDIA. (**c**) The percentage of shared or uniquely identified proteins, using the spectral library or directDIA, that is represented by the specified number of peptides. (**d**) The number of regulated proteins identified using the different library searches and the specified empirical Bayes cut-offs for statistical significance.

**Figure 3 cells-10-01588-f003:**
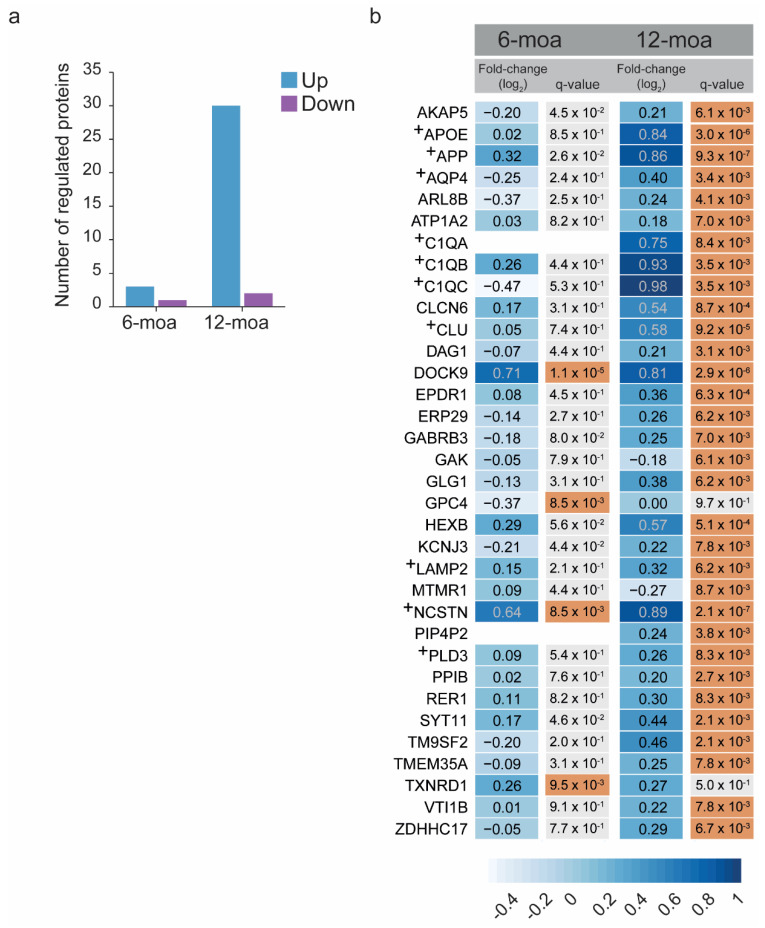
Differential expression analysis of 6- and 12-moa APP/PS1 mice. (**a**) Number of significantly higher and lower expressed proteins (empirical Bayes *q*-value ≤ 0.01) observed at 6- and 12-moa. (**b**) Fold-changes (log_2_) and *q*-values of regulated proteins in at least one of the two datasets. Fold changes are emphasized in color from most extreme decrease (light blue) to highest increase (dark blue). *q*-values ≤ 0.01 are highlighted in orange. Proteins labeled with a+ have been implicated in human AD in previous reports, see Appendix A.

**Figure 4 cells-10-01588-f004:**
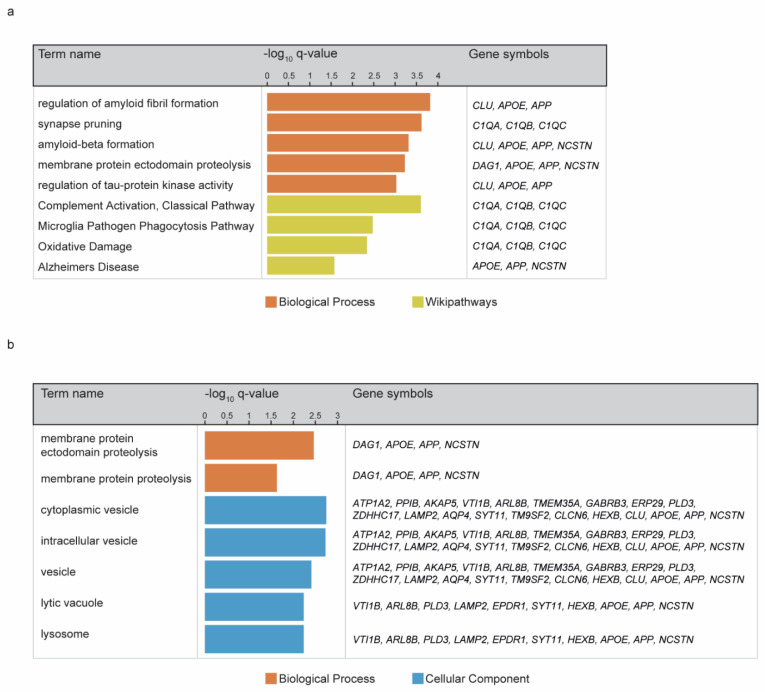
GO enrichment analysis on upregulated proteins in the 12-moa APP/PS1 dataset. (**a**) Enrichment analysis using the whole mouse genome as background results in terms related to Aβ and AD. (**b**) The use of the total list of proteins detected at 12-moa as background results in enrichment terms related to proteolysis and lysosome.

**Figure 5 cells-10-01588-f005:**
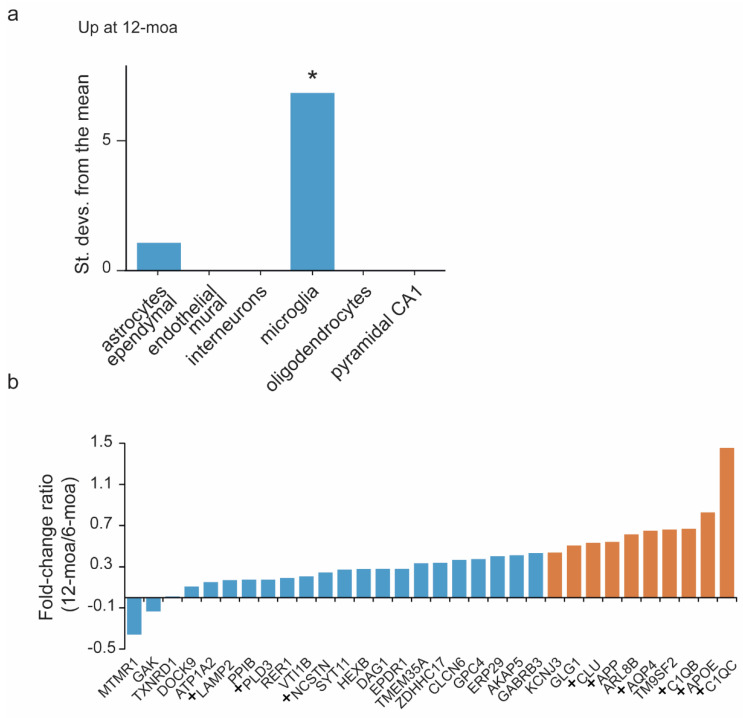
Cell-type enrichment and analysis of changing levels of protein expression over time. (**a**) Expression Weighted Cell-type Enrichment performed using single cell RNAseq level 1 cell-type data obtained from [27], on the upregulated proteins at 12-moa. * *p*-value < 0.01. (**b**) Fold-change ratios are shown for all proteins regulated in at least one of the two datasets demonstrating their level of regulation over time. Proteins labeled with a + have been implicated in human AD in earlier reports, see Appendix A. The top 10 most regulated proteins over time are highlighted in orange.

## Data Availability

All proteomics data used here have been deposited to the ProteomeXchange Consortium via the PRIDE [20] partner repository with the dataset identifier PXD025777.

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
