# Peer review of "Age-Dependent Hippocampal Proteomics in the APP/PS1 Alzheimer Mouse Model: A Comparative Analysis with Classical SWATH/DIA and directDIA Approaches"

_cells, 2021, doi:10.3390/cells10071588_

Round 1

Reviewer 1 Report

The paper by Sophie J.F. van der Spek and colleagues examine the differential protein expression levels in hippocampal synapse enriched fractions obtained from 6 and 12 months APP/PS1 mice in comparison with wild type (wt) controls. The study is very interesting, since AD therapy is still a big challenge, therefore any effort in understanding the underlying mechanisms of this neurodegenerative disorder is highly appreciated. In this light, the idea to unravel temporal changes of key proteins in relation to the advancing Aβ challenge might be of great importance. Overall, the paper is interesting, especially for those who are experts in the proteomics field.

I believe that methods and results are described with high accuracy, and basic information is also provided, so the content could also be understood by those readers who are not experts in this specific field.

I believe that there are two main weaknesses that the authors need to address in order for the paper to be accepted for publication in Cells. First, the author mention several proteins, and in mice of 12 months of age, they highlight differences in the level of specific proteins (e.g. PLD3, VTI1B, EPDR1, HEXB, ARL8B, LAMP2), however they do not specify their specif role and function within cells. This issue makes the text difficult to read. Second, the authors do not offer a clear speculation on the possible role of the deregulated proteins.

Reviewer 2 Report

The authors analyze proteomic changes in an amyloidosis mouse model APP/PS1 using classical and direct DIA approaches. As synaptic dysfunction is at the base of Alzheimer´s pathology, it is of advantage that synaptosomal fractions have been selected for the proteomic analysis. However, it is surprising that relatively little enrichment for synaptic proteins has been observed in this study. Moreover, proteomic alterations were only detectable at late stages of amyloid deposition.

Increasing number of proteomic studies in AD models and human brains has already been reported, compromising the novelty of this study. Moreover, previously reported proteomic studies performed in whole brain tissue or isolated brain cells of AD models or human brains have not been acknowledged and discussed. The relevance of the identified datasets has not been discussed in light of current literature to facilitate critical interpretation of the reported findings. Further specific comments are as follows:

  • The authors did not validate any of the identified hits using other approaches (IHC or biochemistry). Synaptosomal enrichment has not been demonstrated. Were the mice included into the study sex-balanced (sex should be reported)?
  • The authors detected overrepresentation of microglial proteins in their datasets. Taking into account synaptic enrichments, one would have expected increased coverage of neuronal and synaptic proteins.
  • The authors propose that alterations in APOE are a consequence of a long-term exposure to amyloid beta pathology. However, APOE upregulation in AD models has been demonstrated to occur in microglia and transcriptomic and proteomic studies of microglia (such as Sala Frigerio et al., 2019: The Major Risk Factors for Alzheimer's Disease: Age, Sex, and Genes Modulate the Microglia Response to Aβ Plaques - PubMed (nih.gov) and Parhizkar et al., 2019: Loss of TREM2 function increases amyloid seeding but reduces plaque-associated ApoE - PubMed (nih.gov)) detected upregulation of APOE among the earliest pathological changes. Study by Sebastian Monasor et al., 2020 (Fibrillar Aβ triggers microglial proteome alterations and dysfunction in Alzheimer mouse models - PubMed (nih.gov)) reported proteomic alterations of APOE in microglia isolated from two distinct amyloidosis mouse models already at 3-moa. Thus, immune alterations, including upregulation of APOE, are early pathological changes in AD, and not as the authors suggested “late phase microglial activation” (line 465-467). Reasons for not detecting early immune alterations despite the enrichment of microglial proteins may be of technical nature and this limitation should be discussed.
  • The authors discuss that lack of proteomic changes at 6-moa may be due to the compensatory mechanism while 3-moa may represent early responses and 12-moa late responses to amyloid pathology. As the authors anticipated (“The cause of AD is unknown, however, the toxic effects of amyloid-β (Aβ) are believed to play a role in its onset. To investigate this, we examined… “; lines 11-14) the major aim of the current study was to discover early pathological changes. Thus, 3-moa would have been a good selection for this analysis. Nevertheless, it is unlikely that all early changes are compensated and not detectable any longer at 6-moa. Many of the early proteomic changes, at least in microglia (Sebastian Monasor et al., 2020), are detectable and even more pronounced at 6 and 12-moa, supporting a gradual and progressive response to amyloid beta pathology.
  • The authors should discuss their data in light of recently published proteomic study by Li et al., 2021 (Sequence of proteome profiles in preclinical and symptomatic Alzheimer's disease - PubMed (nih.gov)) that reported proteomic changes in synaptic homeostasis in AD patients, further supporting the relevance of this pathway for disease pathology.

Reviewer 3 Report

Major points:

1) In the Introduction, the authors point to the hippocampus as the area of origin of AD pathology (line 31) and refer to Ref.#3 as the source of this information. Contrary to the authors’ claim, this paper does not say that AD originates in the hippocampus. The hippocampus is only mentioned in Table 1 of Ref.#3 in relation to hippocampal sclerosis as one of the common causes of dementia and associated characteristics.

2) In the Introduction, please refer to Raskin et al. (2015) and discuss your choice of tissue in light of their statement: “Beta-amyloid deposition in brain follows a distinct spatial progression starting in the basal neocortex, spreading throughout the hippocampus, and eventually spreading to the rest of the cortex.” (Raskin J, Cummings J, Hardy J, Schuh K, Dean RA. Neurobiology of Alzheimer's Disease: Integrated Molecular, Physiological, Anatomical, Biomarker, and Cognitive Dimensions. Curr Alzheimer Res. 2015;12(8):712-22. doi: 10.2174/1567205012666150701103107.)

3) The rationale why this study was carried out on hippocampal tissue should be further discussed (benefits over the use of frontal cortex, for example).

4) In another study (Zhou et al., 2018), a triple transgenic mouse model for AD was used to measure the effect of donezepil on the proteomic landscape of the hippocampus (because of the known association between this brain area and cognitive functions). Please discuss the findings of your study in relation to that of Zhou et al. (2018) (Zhou X, Xiao W, Su Z, Cheng J, Zheng C, Zhang Z, Wang Y, Wang L, Xu B, Li S, Yang X, Pui Man Hoi M. Hippocampal Proteomic Alteration in Triple Transgenic Mouse Model of Alzheimer's Disease and Implication of PINK 1 Regulation in Donepezil Treatment. J Proteome Res. 2019 Apr 5;18(4):1542-1552. doi: 10.1021/acs.jproteome.8b00818.)

5) Did the identified proteins include post-translationally modified proteins?

6) Please include details on how the tissue was homogenized (weight, volume, etc.).

Minor points:

7) As Spectronaut is an emerging platform for supporting new mass spectrometry technologies, it could be beneficial to the readers to describe its use in a more detailed way (e.g., give more details than “default settings” in line 112)

8) Please provide a list of abbreviations as many of them were never spelled out fully (e.g., SWATH, FASTA, LC-MS, MMTS, etc.)

9) The names and addresses of companies should be included

10) Line 30: secretases?

11) Line 85: 22 microg?

12) Line 88: uL should me microL?

13) Line 264: closing parenthesis is missing

14) In general, figures are hard to read. It is especially true for Figures 1, 2, 3 and 5, as well as Supplementary Figures 1, 2, 3 6, 7 and 8. Please enlarge them/make the white space withing the panels smaller, use full width, etc.)

15) Supplementary Tables 1 and 2 are in graphical format, they may also be difficult to read in the final version.

Round 2

Reviewer 2 Report

The authors addressed some of the raised criticism and largly improved the discussion. Remaining comments could be considered:

1) APP-mouse models (line 430) should be replaced by amyloidosis (or AD) mouse models as the referenced model overexpresses both APP and PS1 (L166P). In additon, reference #35 also uses APPswePS1deltaE9 mice so the authors should be more carefully with mouse model specifications, such us along the lines 430 and 433/434 (it would be helpful to specify the APPPS1 model used according to their mutations or include citation to avoid misunderstandings regarding the PS1 mutation). In addition it should be pointed out when the same mouse model is examined as in the current study.

2) Line 439: Replace APP-models with amyloidosis (or AD) mouse models.

3) Line 451/452 indicates "Upregulation of NCSTN implicates increased levels of γ-secretase...". As the authors pointed out, γ-secretase is a multimeric protein complex and it has been demonstrated that concomitant increase in all subunits is necessary to confer the increased activity. Thus, upregulation of NCSTN alone would not be sufficient to hypothesize increased γ-secretase activity.
